# A systematic review and meta-analysis of eyespot anti-predator mechanisms

Ayumi Mizuno[1,2,3]*, Malgorzata Lagisz[2], Pietro Pollo[2†], Yefeng Yang[2†], Masayo Soma[1], Shinichi Nakagawa[2,3,4]

[1]Department of Biology, Faculty of Science, Hokkaido University, Sapporo, Japan; [2]Evolution & Ecology Research Centre, School of Biological, Earth and Environmental Sciences, The University of New South Wales, Sydney, Australia; [3]Department of Biological Sciences, Faculty of Science, The University of Alberta, Edmonton, Canada; [4]Theoretical Sciences Visiting Program, Okinawa Institute of Science and Technology Graduate University, Onna, Japan

## eLife assessment

This meta-analysis presents **valuable** findings that reexamine the function of butterfly eyespots in predator avoidance and report for conspicuousness over mimicry. The analysis is robust, but the evidence supporting the importance of conspicuousness is **incomplete** due to the limitations of the literature, and this debate would benefit from additional experiments that would strengthen these claims. This paper is of interest to evolutionary biologists and ecologists working on the evolution of morphology and predator-prey interactions.

**\*For correspondence:**
ayumi.mizuno5@gmail.com

†These authors contributed equally to this work

**Competing interest:** The authors declare that no competing interests exist.

**Abstract** Eyespot patterns have evolved in many prey species. These patterns were traditionally explained by the eye mimicry hypothesis, which proposes that eyespots resembling vertebrate eyes function as predator avoidance. However, it is possible that eyespots do not mimic eyes: according to the conspicuousness hypothesis, eyespots are just one form of vivid signals where only conspicuousness matters. They might work simply through neophobia or unfamiliarity, without necessarily implying aposematism or the unprofitability to potential predators. To test these hypotheses and explore factors influencing predators' responses, we conducted a meta-analysis with 33 empirical papers that focused on bird responses to both real lepidopterans and artificial targets with conspicuous patterns (i.e. eyespots and non-eyespots). Supporting the latter hypothesis, the results showed no clear difference in predator avoidance efficacy between eyespots and non-eyespots. When comparing geometric pattern characteristics, bigger pattern sizes and smaller numbers of patterns were more effective in preventing avian predation. This finding indicates that single concentric patterns have stronger deterring effects than paired ones. Taken together, our study supports the conspicuousness hypothesis more than the eye mimicry hypothesis. Due to the number and species coverage of published studies so far, the generalisability of our conclusion may be limited. The findings highlight that pattern conspicuousness is key to eliciting avian avoidance responses, shedding a different light on this classic example of signal evolution.

## Introduction

Naturalists have long pondered the evolution and function of the many signals and cues animals use to communicate (*Endler, 1992*; *Endler, 1993*; *Andersson, 1994*; *Johnstone, 1996*; *Martin Schaefer et al., 2004*; *Johansson and Jones, 2007*; *Hill, 2009*; *Jones and Ratterman, 2009*; *Rose et al., 2022*). Visual signals, such as vibrant colours and contrasting patterns, have attracted more interest

**Figure 1.** A visual summary of three hypotheses that explain the predation avoidance function of eyespot patterns and the predictions that can be derived from these two hypotheses. The resemblance of eyespots to actual eyes is discussed through the predator mimicry hypothesis and the conspicuous signal hypothesis. The table shows the predictions derived from these two hypotheses. The references of the examples illustrated in the figure: cuckoos and hawks (*Davies and Welbergen, 2008*; *Ma et al., 2018*); moths and spiders (*Rota and Wagner, 2006*); poison frogs (*Saporito et al., 2007*); ladybugs (*María Arenas et al., 2015*); plovers (*de Framond et al., 2022*); lizards (*Bateman and Fleming, 2009*).

from researchers than other signals, likely because our species is visually oriented (*Endler, 1992*; *Kelber et al., 2003*; *Endler et al., 2005*). Eyespot patterns, characterised by concentric rings of different colours with a light outer ring and a dark centre (*Stevens, 2005*), are well-known patterns believed to reduce predation. Although eyespots have been researched for a long time (*Stevens, 2005*; *Kodandaramaiah, 2011*; *Stevens and Ruxton, 2014*; *Drinkwater et al., 2022*), researchers continue to debate why eyespots might deter predation.

Three hypotheses have been proposed to explain why eyespot patterns can contribute to prey survival (reviewed in *Stevens, 2005*; *Kodandaramaiah, 2011*; *Stevens and Ruxton, 2014*; *Figure 1*). First, the eye mimicry hypothesis suggests that eyespots play a role in deterring predators from attacking prey and reducing predation risks by mimicking the eyes of vertebrates (*Blest, 1957a*; *Vallin et al., 2005*; *Kjernsmo and Merilaita, 2017*). This hypothesis predicts that if the pattern has specific

characteristics (e.g. eye-like shape) and is presented as a pair, predation avoidance will increase, assuming eyespots imitate potential predators. Second, the conspicuousness hypothesis posits that eyespots are simply conspicuous patterns that prevent attacks due to negative predator responses caused by sensory bias, neophobia, or sensory overload (*Stevens, 2005*; *Stevens and Ruxton, 2014*). The hypothesis states that the eye-like shape and patterns arranged in pairs do not necessarily deter predators. Rather, it is their conspicuous appearance that makes them effective predator deterrents, and any resemblance to eyes is coincidental. Eyespots can act as an aposematic signal for potential predators. For example, if the size of the pattern (one of the measures of conspicuousness) increases, the avoidance effect will also increase. Third, the deflection hypothesis suggests that predator attacks should be directed toward eyespots to avoid damage to vital body parts (*Hill and Vaca, 2004*; *Olofsson et al., 2010*; *Kodandaramaiah et al., 2013*; *Olofsson et al., 2013a*; *Merilaita et al., 2017*). The eye mimicry and conspicuousness hypotheses are usually applied to explain large eyespots, while the deflection hypothesis is used to interpret the function of small ones (*Stevens, 2005*; *Kodandaramaiah, 2011*; *Stevens and Ruxton, 2014*). The first two of these hypotheses focus on how eyespots prevent predators from attacking, specifically whether it is because they resemble eyes or are conspicuous. The third hypothesis focuses on whether eyespots divert a predator's attack away from vital body parts by drawing the predator's attention to them. Thus, in this third hypothesis, whether the eyespots resemble eyes or are conspicuous is not the central issue (*Stevens, 2005*; *Kodandaramaiah, 2011*; *Stevens and Ruxton, 2014*). Although there seems to be little disagreement in the deflection hypothesis (*Lyytinen et al., 2004*; *Pinheiro et al., 2014*; *Ho et al., 2016*, but see also *Lyytinen et al., 2003*), why large eyespots can intimidate avian predators has been controversial (*Stevens, 2005*; *Stevens and Ruxton, 2014*). This is because while the eye mimicry and conspicuousness hypotheses are not mutually exclusive, the key mechanism that explains why predators react negatively to eyespots is clearly different.

Lepidopterans, such as butterflies and moths, have been the leading models for testing the eye mimicry and conspicuousness hypotheses. A typical empirical study has adult individuals, caterpillars, or their models as prey, with birds as predators (reviewed in *Stevens, 2005*; *Stevens and Ruxton, 2014*; *Kodandaramaiah, 2011*). According to the eye mimicry hypothesis, avian predators perceive the eyespots as the eyes of a potential enemy. For example, great tits (*Parus major*) showed more aversive responses to animated butterflies with a pair of large eyespots than those without, and such eyespots were more effective than modified, less mimetic, but equally contrasting patterns (*De Bona et al., 2015*). Although several studies have supported the eye mimicry hypothesis (e.g. *Blest, 1957a*; *Merilaita et al., 2011*; *De Bona et al., 2015*), many conspicuous patterns other than eyespots, such as dots and stripes, likely deter attacks from predators as well (*Stevens et al., 2008a*; *Stevens et al., 2009a*; *Dell'aglio et al., 2016*; *Ximenes and Gawryszewski, 2020*). Some field experiments with artificial prey have supported the conspicuousness hypothesis, demonstrating survival rates for both conspicuous (eyespots and non-eyespots) pattern prey stimuli were higher than control prey stimuli (*Stevens et al., 2007b*; *Stevens et al., 2008a*; *Stevens et al., 2009a*). Such discrepancies might have arisen from differences in experimental design between studies, such as the size, number, and shape of the presented pattern stimuli or the bird species used as subjects in the experiments (*Stevens, 2005*; *Stevens, 2007a*). However, there has been no systematic attempt to synthesise and compare earlier studies quantitatively.

Here, we conduct a systematic review with meta-analysis to synthesise empirical evidence on the intimidating effects of eyespots and the factors that contribute to predator avoidance responses towards them. To examine the two hypotheses above, we ask three interrelated questions. First, we examine whether conspicuous patterns, namely eyespots and non-eyespot patterns (i.e. conspicuous patterns other than eyespots), influence bird responses or prey survival in a manner that increases the success of predator avoidance. Second, we test whether pattern resemblance to eyes (eye-like shape) is the key to predator avoidance (which differentiates the eye mimicry hypothesis from the conspicuousness hypothesis). For the first and second questions, we use (phylogenetic) multilevel meta-analytic models. Third, we examine what factors promote bird response and increase prey survival by testing eight moderators (treatment stimulus pattern types, namely eyespots vs. non-eyespots, pattern area, the number of pattern shapes, prey material type, maximum pattern diameter/length, total pattern area, total area of prey surface, and prey shape type) (*Figure 2bc*). For the third question, we apply meta-regression models to evaluate how these moderators influence predator avoidance. We assess

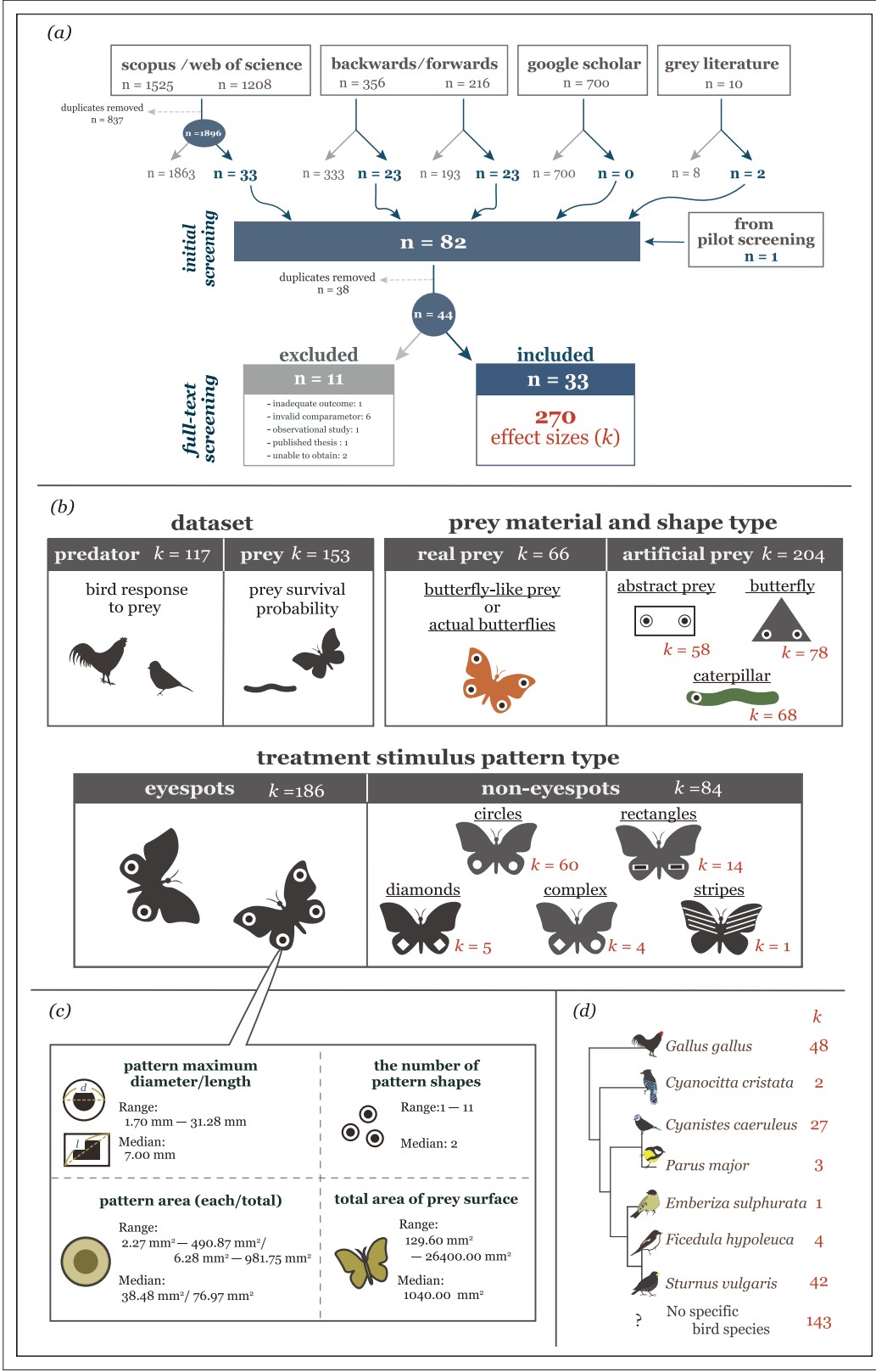

**Figure 2.** Overview of the dataset. (**a**) Preferred Reporting Items for Systematic Reviews and Meta-Analyses (PRISMA)-like flowchart of the systematic literature search for the meta-analysis. (**b**) and (**c**) Details of the main moderators examined in the meta-analysis. (**d**) The phylogenetic tree of bird species included in the meta-analysis, together with the sample sizes and number of effect sizes per species.

publication bias to check the robustness of our findings. Throughout our review and analysis process, we adhere to PRISMA (Preferred Reporting Items for Systematic Reviews and Meta-Analyses; *Moher et al., 2009*) and PRISMA-EcoEvo (Preferred Reporting Items for Systematic reviews and Meta-analyses in Ecology and Evolutionary biology; *O'Dea et al., 2021*) guidelines and report this study (*Figure 2a*; *Supplementary file 1*; for detailed methods, see **Materials and methods**).

## Results

### Screening outcomes and dataset characteristics

We obtained 270 effect sizes from 33 studies (164 experiments) for our analysis (*Blest, 1957a*; *Jones, 1980*; *Inglis et al., 1983*; *Wourms and Wasserman, 1985*; *Lyytinen et al., 2003*; *Forsman and Herrström, 2004*; *Lyytinen et al., 2004*; *Vallin et al., 2005*; *Stevens et al., 2007b*; *Stevens et al., 2008a*; *Stevens et al., 2008b*; *Stevens et al., 2009a*; *Stevens et al., 2009b*; *Brilot et al., 2009*; *Kodandaramaiah et al., 2009*; *Vallin et al., 2010*; *Merilaita et al., 2011*; *Vallin et al., 2011*; *Blut et al., 2012*; *Hossie and Sherratt, 2012*; *Wert, 2012*; *Hossie and Sherratt, 2013*; *Olofsson et al., 2013a*; *Olofsson et al., 2013b*; *Stevens et al., 2013*; *Skelhorn et al., 2014*; *Hossie et al., 2015*; *Mukherjee and Kodandaramaiah, 2015*; *Olofsson et al., 2015*; *Ho et al., 2016*; *Skelhorn et al., 2016*; *Postema, 2022*). The screening process and reasons for exclusion at the full-text screening stage are summarised in the PRISMA-like flowchart (*Figure 2a*), with additional details available in *Supplementary file 2*, which comprises a list of included/excluded studies. Of the dataset, 68.9% of effect sizes came from eyespot presentation experiments (*Figure 2b*). The remaining 31.1% of effect sizes came from non-eyespot pattern presentation experiments (*Figure 2b*). The latter category encompassed various shapes, including circles (71.4%), rectangles (16.7%), diamonds (6.0%), complex patterns (combinations of circles and diamonds; 4.8%), and stripes (1.1%); 93.7% of the control stimuli used in these experiments involved the removal of the pattern used in the treatment stimuli; the remaining stimuli were camouflage patterns (6.3%). Prey shape type used for stimulus presentation varied from real or imitation of a particular butterfly (24.4%) to simply a piece of paper (21.5%) (*Figure 2b*). The number of pattern shapes varied between studies from one to 11, but in

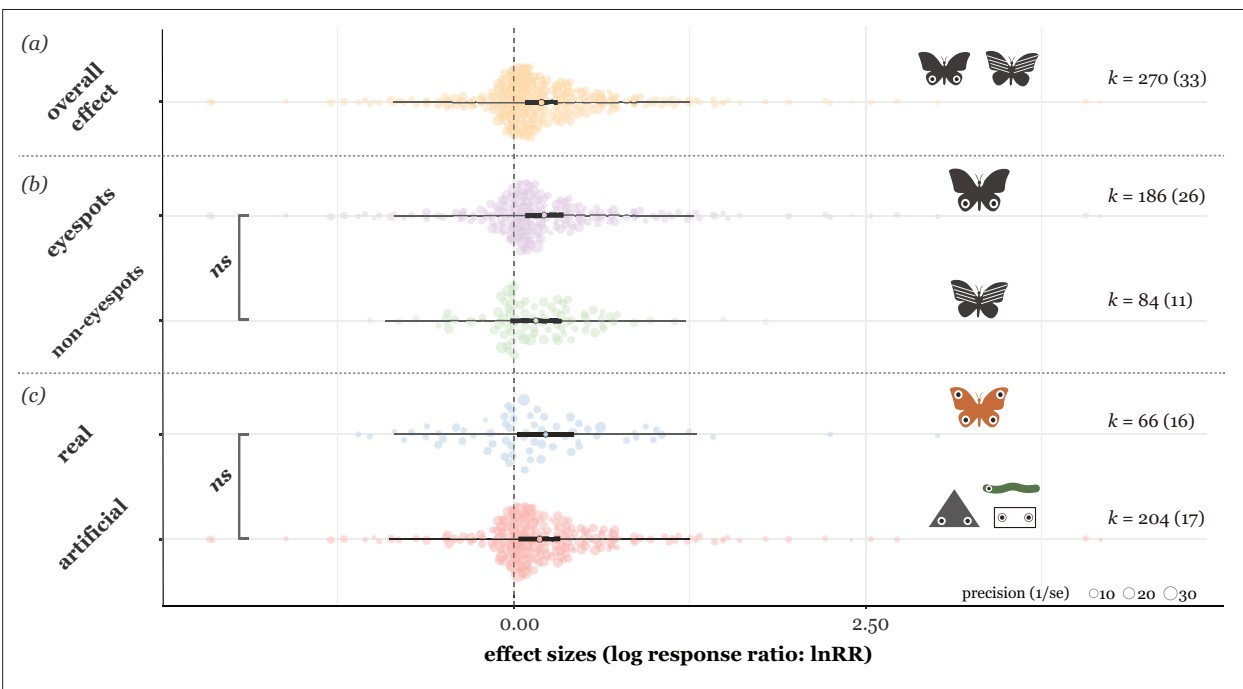

**Figure 3.** Mean effect sizes of (**a**) overall for conspicuous patterns (eyespots and non-eyespots), (**b**) effects split by experiments with eyespot versus non-eyespot patterns, and (**c**) two prey types used in the experiments. Thick horizontal lines represent 95% confidence intervals, and thin horizontal lines represent 95% prediction intervals. The points in the centre of each thick line indicate the average effect size. *k* is the number of effect sizes used to estimate the statistics, followed by the number of studies in the brackets.

most experiments, they were two (i.e. a pair of shapes; *Figure 2c*). Additionally, we found that the size of these patterns, both area and maximum diameter/length, exhibited considerable variation across studies (*Figure 2c*). The total area of the patterns and stimulus also varied widely (*Figure 2c*). The studies reported responses to conspicuous pattern stimuli by seven bird species (*Figure 2d*). Chickens (*Gallus gallus*) and common starlings (*Sturnus vulgaris*) were the most studied birds in our dataset. Apart from chickens (eight studies) and Eurasian blue tits (*Cyanistes caeruleus*; five studies), effect sizes were available from just one or two studies per species. Six of the seven species were omnivores, and one (yellow bunding; *Emberiza sulphurata*) was a granivore (*Tobias et al., 2022*).

## Does the presence of conspicuous patterns affect predator avoidance?

The overall mean effect size, calculated as the natural logarithm of the response ratio (lnRR) in this study, was statistically significant (for details on effect size calculation, see **Materials and methods**). This showed a 21.86% (the percentage value is the back-transformed values of lnRR) increase in the probability of predator avoidance, such as higher prey survival rates or eliciting fewer attacks from birds (estimate=0.20, 95% CI = [0.08, 0.31], $t_{[df = 268]}$ = 3.40, p = 0.0008), in prey with conspicuous patterns than in prey without such patterns (*Figure 3a*). Total heterogeneity across effect sizes was high ($I^2$=96.50%); more specifically, observation ID (representing the within-study effect) accounted for the most heterogeneity, 79.88%, with study ID (representing between-study effect) accounting for the remaining 16.61%.

## Is there a difference in predator avoidance between eyespots and non-eyespot patterns?

There was no statistically significant difference between the effects of eyespots and non-eyespot patterns ($F_{[df1 = 1, df2 = 268]}$ = 0.33, p = 0.57, $R^2$ = 0.27%; *Figure 3b*). On average, eyespot patterns resulted in 24.37% (estimate = 0.22, 95% CI = [0.08, 0.35], $t_{[df = 268]}$ = 3.17, p = 0.002) and non-eyespot patterns in 17.11% (estimate = 0.16, 95% CI = [–0.02, 0.34], $t_{[df = 268]}$ = 1.71, p = 0.09) increases in predator avoidance compared with control stimuli, although this trend was not statistically significant for non-eyespots (*Figure 3b*).

## What factors promote predator avoidance?

Our uni-moderator meta-regression model with pattern area (individual shape area) showed that larger patterns were associated with an increase in predator avoidance (estimate = 0.11, 95% CI = [0.03, 0.19], $t_{[df = 268]}$ = 2.71, p = 0.007, $R^2$ = 8.56%; *Figure 4a*). The total pattern area also promoted predator avoidance (estimate = 0.09, 95% CI = [0.004, 0.17], $t_{[df = 268]}$ = 2.07, p = 0.04, $R^2$ = 5.18%; *Figure 5a*). Similarly, the maximum diameter/length of the pattern positively influenced predator avoidance (estimate = 0.19, 95% CI = [0.04, 0.35], $t_{[df = 268]}$ = 2.46, p = 0.01, $R^2$ = 6.62%; *Figure 5b*). In contrast, an increased number of pattern shapes significantly reduced the effect of predator avoidance (estimate = –0.06, 95% CI = [-0.11, –0.008], $t_{[df = 268]}$ = –2.29, p = 0.02, $R^2$ = 2.46%; *Figure 4b*). We found no significant effects of total prey surface area on predator avoidance (estimate = –0.03, 95% CI = [–0.15, 0.09], $t_{[df = 268]}$ = –0.48, p = 0.63, $R^2$ = 0.42%; *Figure 5c*). Predator avoidance was not statistically significantly affected by differences in whether the presented prey looked like a real lepidopteran species ($F_{[df1 = 1, df2 = 268]}$ = 0.12, p = 0.72, $R^2$ = 0.13%). Both types of prey material (real/imitation and abstract butterfly) had similar positive trends (*Figure 3c*), with the former increasing predator avoidance by 25.55% (estimate = 0.23, 95% CI = [0.03, 0.43], $t_{[df = 268]}$ = 2.24, p = 0.03) and the latter by 20.07% (estimate = 0.18, 95% CI =[0.04, 0.33], $t_{[df = 268]}$ = 2.44, p = 0.02). Furthermore, when also considering prey type (*Figure 6*), abstract and real butterflies significantly exhibited increased predator avoidance by 37.98% (estimate = 0.32, 95% CI = [0.11, 0.53], $t_{[df = 268]}$ = 3.04, p = 0.003) and by 25.40% (estimate = 0.23, 95% CI = [0.03, 0.42], $t_{[df = 268]}$ = 2.25, p = 0.03), respectively, but artificial abstract caterpillars (estimate = 0.07, 95% CI = [–0.18, 0.31], $t_{[df = 266]}$ = 0.53, p = 0.60) and artificial abstract prey (estimate = 0.01, 95% CI = [–0.35, 0.37], $t_{[df = 266]}$ = 0.06, p = 0.95) did not, respectively. When comparing each prey type (e.g. abstract butterfly vs. real butterfly), none of the differences was statistically significant (*Figure 6*).

The multi-moderator (full) regression model showed that only pattern area positively affected predator avoidance (estimate = 0.10, 95% CI = [0.009, 0.18], $t_{[df = 266]}$ = 2.16, p = 0.03; *Supplementary file 3*). Contrary to the uni-moderator regression model, the number of patterns showed no significant

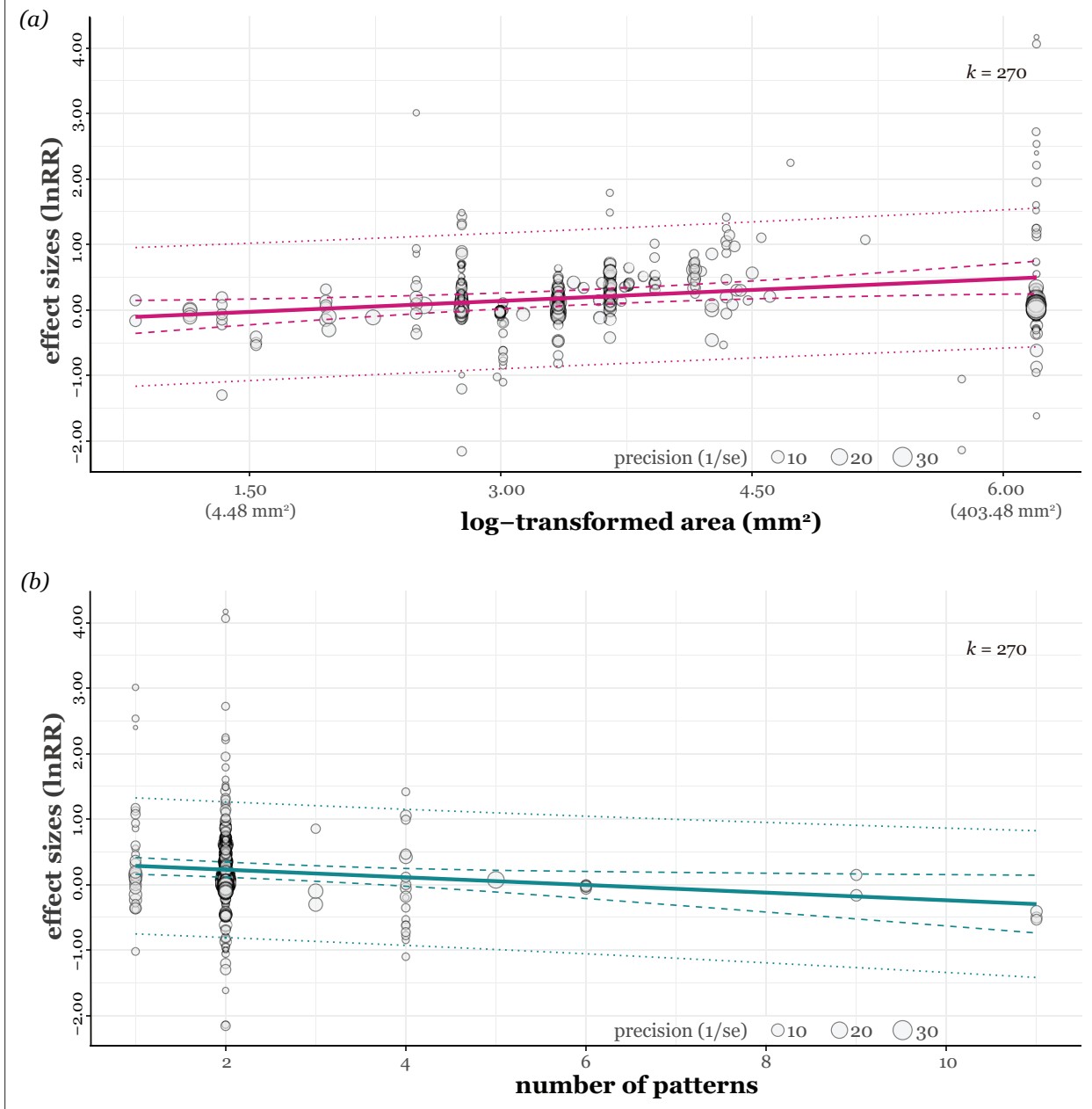

**Figure 4.** The relationships between (**a**) prey conspicuous pattern area (log-transformed) and effect sizes and (**b**) number of prey conspicuous patterns and effect sizes. Circle sizes are scaled according to precision, *k* represents the number of effect sizes. Each fitted regression line is shown as a coloured straight line, and 95% confidence and prediction intervals are shown as dashed and dotted coloured lines, respectively.

effects on predator avoidance, although the consistent trend remained (estimate = –0.05, 95% CI = [–0.11, 0.004], $t_{[df = 266]}$ = –1.84, p = 0.07; *Supplementary file 3*). The full model accounted for 8.33% of the variation in the dataset. The complete output of the multi-moderator model is displayed in *Supplementary file 3*.

## Publication bias

The funnel plot showed no visual sign of funnel asymmetry (*Figure 7a*). The meta-regression analysis, which included the square root of the inverse of the effective sample size, further supported this observation by showing that the effective sample size did not significantly predict the effect size values (estimate = –0.09, 95% CI = [–0.83, 0.65], $t_{[df = 266]}$ = –0.24, p = 0.81; *Figure 7b*). There was

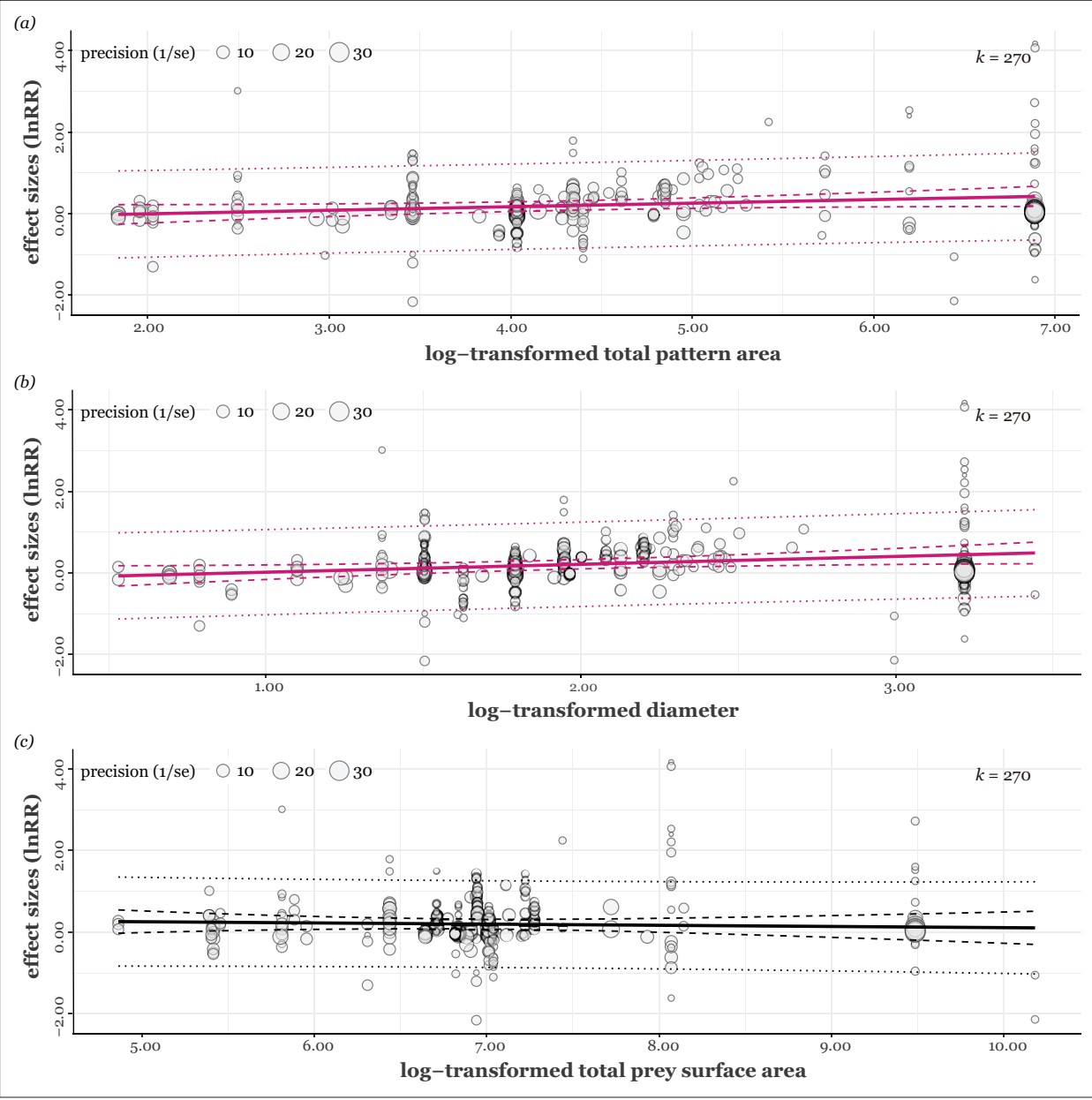

**Figure 5.** The relationships between (**a**) total pattern area, (**b**) pattern maximum diameter/length, and (**c**) total prey surface area and effect sizes. $k$ shows the number of effect sizes. Each fitted regression line is shown as a solid straight line, and 95% confidence and prediction intervals are shown as dashed and dotted lines, respectively.

no detectable trend suggesting that more recent publications consistently showed lower or higher effect size values, which would have indicated the presence of time-lag publication bias (estimate = −0.0008; 95% CI = [−0.01, 0.01], $t_{[df = 266]}$ = −0.12, p = 0.90; *Figure 7c*). We obtained the same trends from multi-moderator meta-regressions (*Figure 8*).

## Discussion

Eyespots and non-eyespot patterns did not differ significantly in the magnitude of deterring effects (*Figure 3b*). Avian predators showed similar avoidance responses to the conspicuous patterns compared to control ones (*Figure 3a*). Specifically, larger pattern sizes played a crucial role in eliciting negative responses from birds (*Figure 4a*). Furthermore, negative responses from birds showed the tendency to decline with increasing pattern number: single patterns were likely more intimidating

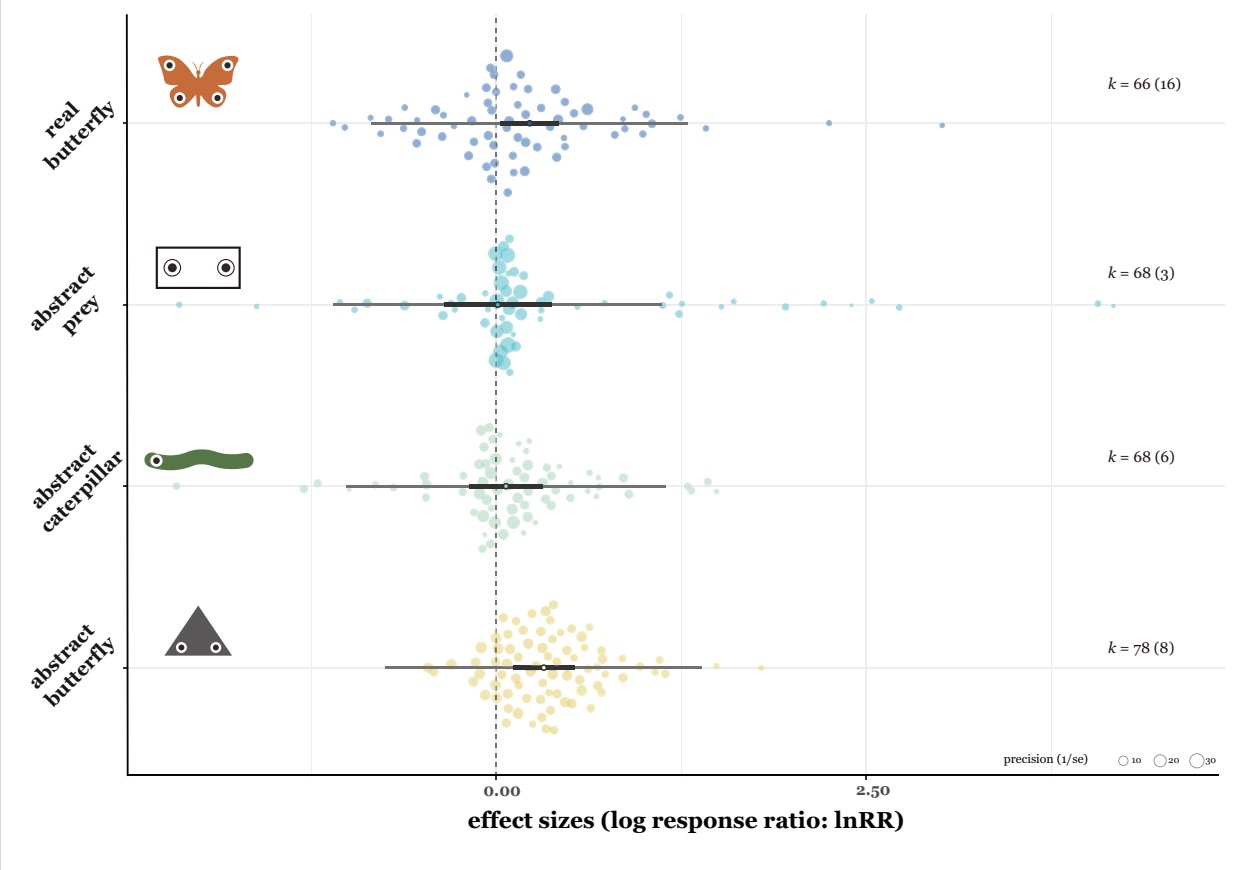

**Figure 6.** Mean effect sizes of total prey shape types. Thick horizontal lines represent 95% confidence intervals, and thin horizontal lines represent prediction intervals. The points in the centre of each thick line indicate the average effect size. *k* shows the number of effect sizes.

than a group of patterns (*Figure 4b*). Taken together, our results support the conspicuousness hypothesis rather than the eye mimicry hypothesis.

## Eye mimicry or conspicuousness hypothesis?

Overall, our meta-analysis showed that conspicuous patterns could increase predator avoidance by over 20%. Specifically, our results indicate that conspicuousness per se can be advantageous in avoiding bird predation (*Figures 3ab and 4*). The evidence favouring the conspicuousness hypothesis comes mainly from a series of field experiments by Stevens and his colleagues (*Stevens et al., 2007b*; *Stevens et al., 2008a*; *Stevens et al., 2009a*). They showed that both eyespots and non-eyespots improved the prey survival similarly compared to non-conspicuous patterns (*Stevens et al., 2007b*; *Stevens et al., 2008a*; *Stevens et al., 2009a*). In addition, their research showed prey with more conspicuous patterns (i.e. large-size patterns) tended to survive more than others (*Stevens et al., 2007b*; *Stevens et al., 2008a*; *Stevens et al., 2009a*), and eye resemblance (e.g. number or pattern shapes) did not significantly affect the prey's survival (*Stevens et al., 2007b*; *Stevens et al., 2008a*; *Stevens et al., 2009a*). Given that these pattern stimuli used in the experiments are rarely or never found in natural environments (*Stevens et al., 2007b*), the most parsimonious explanation for these results is neophobia or dietary conservatism in birds (*Ord et al., 2021*; *Marples et al., 1998*; *Marples and Kelly, 1999*). Both phenomena appear to diminish with habituation and/or learning. A few studies investigated such factors for intimidating effects, and they showed that repeated encounters made birds more habituated to eyespot patterns (*Blest, 1957a*; *Inglis et al., 1983*; *Skelhorn et al., 2014*). We need more systematic tests of bird habituation to vividly- or aposematic-coloured patterns to better understand the evolution and function of such patterns in Lepidoptera.

While our meta-analytic results favour the conspicuousness hypothesis, several empirical studies support the eye mimicry hypothesis. For example, *De Bona et al., 2015* found that a pair of eyespots

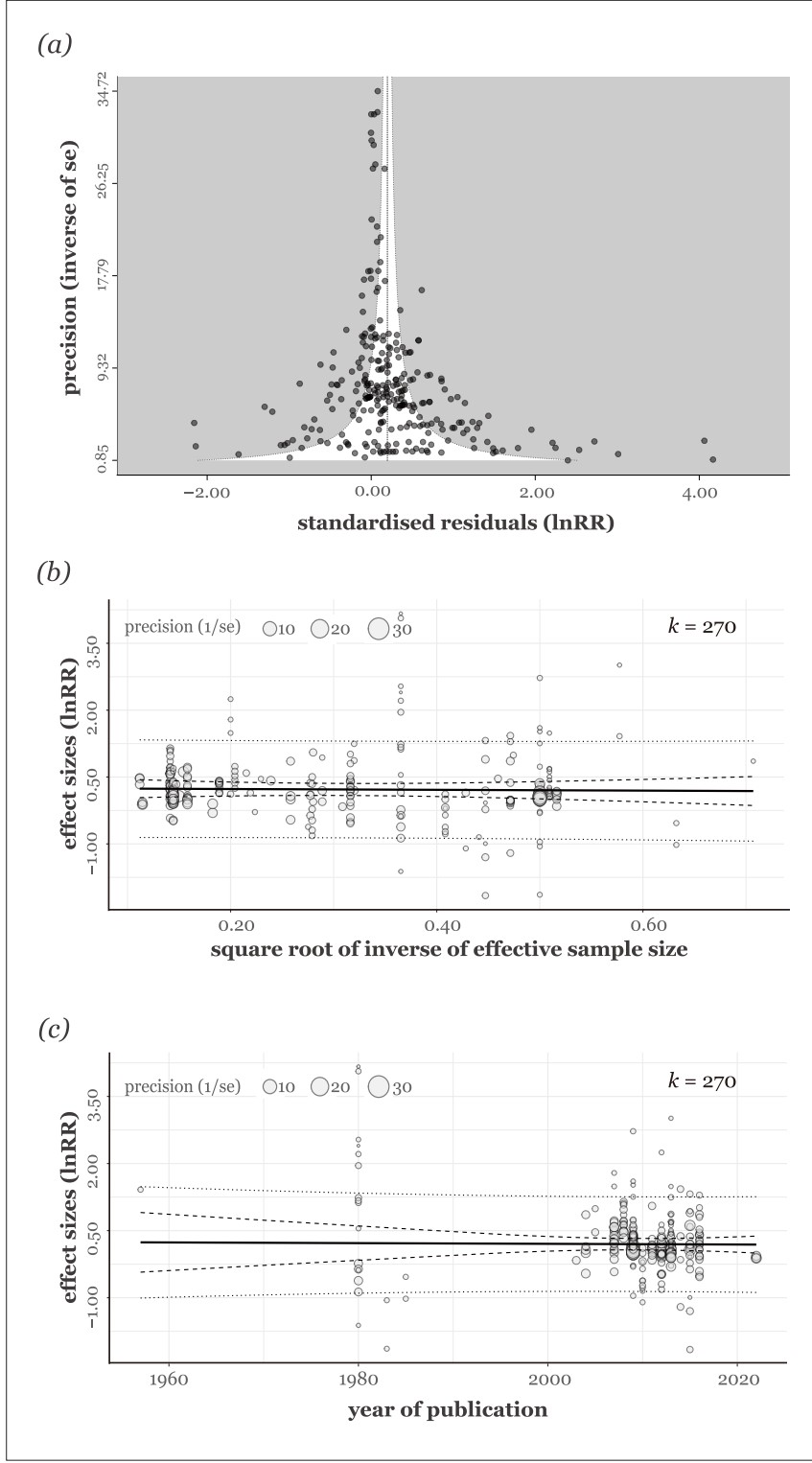

**Figure 7.** Funnel plot and relationships between effect Sizes, effective sample size, and publication year. (**a**) Funnel plot using effect size and its inverse standard error. The relationship between effect sizes and (**b**) the square root of the inverse of effective sample size and (**c**) publication year. In (**b**) and (**c**), circle sizes are scaled accordingly to precision, and $k$ represents the number of effect sizes. Each fitted regression line is shown as a straight line, and 95% confidence and prediction intervals are shown as dashed and dotted lines, respectively.

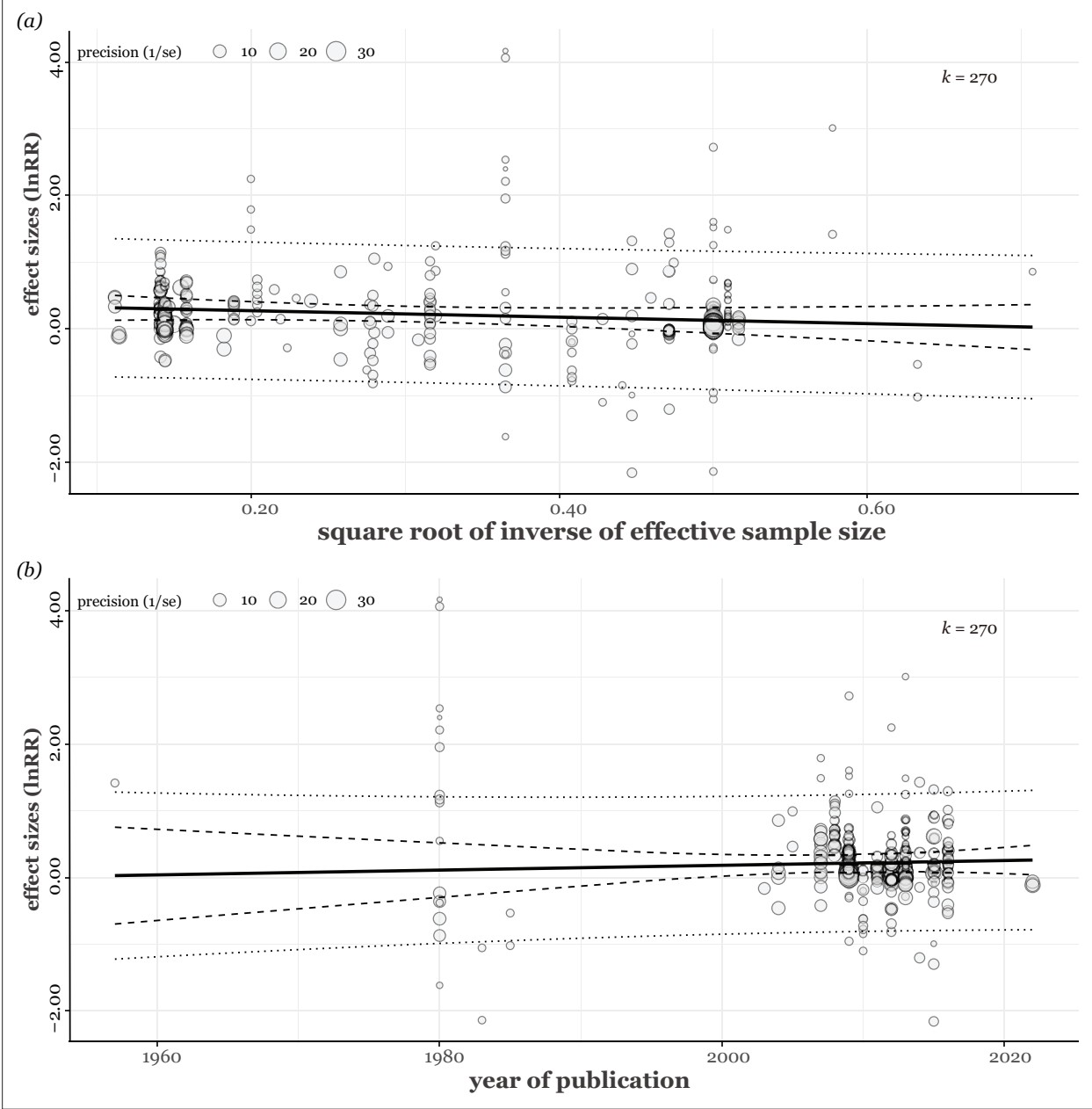

**Figure 8.** The relationship between (**a**) effect sizes and the square root of the inverse of effective sample size and (**b**) relationship between effect sizes and publication year. Both plots were based on the multi-moderator model. *k* shows the number of effect sizes. Each fitted regression line is shown as a solid straight line, and 95% confidence intervals and prediction intervals are shown as dashed and dotted lines, respectively.

of *Caligo martia* was as effective as true owl eyes and more efficient in eliciting predator avoidance responses than less mimetic but equally contrasting circles. ***Blut and Lunau, 2015*** created artificial eye-spotted prey with different similarities to the vertebrate eyes and checked their survival rates in a field experiment. They revealed that the prey with the most mimetic pattern had the highest survival rate (***Blut and Lunau, 2015***). Although studies on Lepidoptera larvae are relatively limited, caterpillar eyespots are considered part of snake mimicry (***Stevens and Ruxton, 2014***). Some research examined the benefit of eyespots by presenting artificial caterpillars (marked with eyespots and control) made from dyed pastry to wild birds and showed that eyespots improved survival (***Hossie and Sherratt, 2012***; ***Hossie and Sherratt, 2013***; ***Skelhorn et al., 2014***). Despite these convincing pieces of empirical evidence, our meta-analytic results showed that eye resemblance did not improve predator

avoidance. If the eye mimic hypothesis was true, we would have seen a clear difference between studies investigating eyespots and non-eyespots.

However, we observed little heterogeneity among studies, despite finding high heterogeneity within individual studies. This finding implies that if each study followed similar experimental procedures within studies, our main result on predator avoidance would be more generalisable. The high within-study heterogeneity can be caused by varying stimulus characteristics contributing to the effect size variations, even in the same studies. Bird phylogenetic relatedness explained little heterogeneity in our predator dataset, but this may have occurred because a limited number of subject bird species (i.e. chickens, common starlings, Eurasian blue tits) dominated our dataset (*Figure 2d*). While we cannot exclude the possibility of species differences in birds' responses to the conspicuous patterns, our analysis indicated that bird species identity did not explain the observed variation in predator avoidance.

We also note that conspicuous patterns can also be important for conspecific communication in butterflies, not just for avoiding predation (*Stevens, 2005*; *Crees et al., 2021*). For example, eyespots on *Bicyclus anynana* are known to function as sexual signals. For example, males choose females depending on eyespot size and reflectance (*Robertson and Monteiro, 2005*). Regarding the non-eyespot patterns, males of *Heliconius cydno* and *H. pachinus* can recognise conspecific females by the bright colour of wing patches (*Kronforst et al., 2006*; *Finkbeiner et al., 2014*). Conspicuous patterns can also act as social signals in other taxa (e.g. birds: *Mason and Bowie, 2020*), but this function remains unclear in butterflies. Therefore, the diversity of patterns on wings could be shaped by intra-specific and inter-specific communication. We should simultaneously consider the influence of anti-predator and sexual/social signalling functions on the evolution of butterfly conspicuous patterns (*Robertson and Monteiro, 2005*; *Ng et al., 2017*; *Huq et al., 2019*).

## What factors explain the observed heterogeneity?

The indicators of pattern size, including each pattern area (*Figure 4a*), total pattern area (*Figure 5a*), and maximum diameter/length (*Figure 5b*), were the most important moderators of effect sizes, overall indicating that large patterns could promote predator avoidance. Notably, these size metrics were correlated, so they are not independent of each other. Several studies suggested that the pattern size difference is related to the difference in prey survival (*Stevens et al., 2008a*; *Kodandaramaiah et al., 2013*; *Ho et al., 2016*). For example, eyespots larger than 6.0 mm may have a strong deterrent effect with increasing size (*Ho et al., 2016*), but such patterns may increase the visibility of lepidopterans, and their presence may increase predation rates as well (*Lindström et al., 2001*). Indeed, small conspicuous patterns tend to attract predators' attention, as explained by the deflection hypothesis (*Stevens, 2005*; *Humphreys and Ruxton, 2018*). The effect may contribute to the observed negative overall effect sizes (*Figures 3 and 4*). Considering studies on *B. anynana* with eyespots with a deflecting effect (maximum diameter is about 5.0 mm; *Supplementary file 5*), a size of at least 6.0 mm is required to avoid predator approach. However, it is uncertain whether the effect would linearly increase with size or whether an optimal size exists. Although eyespot sizes on actual Lepidoptera may be restricted by their body or wing size (e.g. *Hossie et al., 2015*, but see also *Kodandaramaiah et al., 2013*), it would be interesting to find a maximum threshold for patterns that promote predator avoidance responses in birds.

Among other moderators tested (prey material type, total pattern area, and prey shape type), the only moderator that seemed to explain heterogeneity was the number of patterns (*Figure 4b*; yet it is likely inconclusive; see *Supplementary file 3*). Previous studies predominantly employed a single pattern or a pair of patterns, leading to limited variations. Nonetheless, our findings indicate that a single eyespot is equally or more effective than a pair of eyespots. Consequently, the resemblance to a pair of eyes, a crucial aspect of the eye mimicry hypothesis, may be optional for effective predator avoidance. Indeed, we should note that the presence of both eyes is unnecessary for birds to recognise their predators because birds may often see only one eye of their predators. To disentangle the two hypotheses, we recommend conducting the following experiments with two key features (*Stevens, 2007a*; *Stevens et al., 2008a*; *Stevens et al., 2009b*): a set of stimuli that (1) have the same size (area or diameter/maximum length of each pattern or total pattern area) but with different numbers of patterns ranging from a few usually found in Lepidoptera to numerous patterns unlike those seen in them, and (2) are presented with the same number of patterns and the same size

but different pattern shapes. Results from these experiments could deepen our current knowledge, allowing us to inch toward a more definitive answer.

## Knowledge gaps and future opportunities

Along with other conspicuous patterns, eyespots are believed to deter bird predation, and our meta-analysis supports this function. However, five major gaps remain in the current literature and our knowledge. First, birds and humans likely perceive eye-like shapes differently based on the interspecific diversity of bird vision (*Martin, 2017*). For example, most bird species can detect ultraviolet light, which is invisible to humans, and the ultraviolet reflection of the butterflies' eyespots may contribute to predator avoidance (e.g. *Olofsson et al., 2010*, *Olofsson et al., 2013a*). In addition, researchers can quantify and objectively evaluate conspicuousness, such as size and number, but the assessment of 'eye mimicry' remains subjective. Thus, it could be premature to conclude that eyespots on Lepidoptera resemble vertebrate eyes universally.

Second, some lepidopterans present conspicuous patterns to potential predators in combination with other elements, such as sounds and movements (*Blest, 1957a*; *Blest, 1957b*; *Bura et al., 2016*; *Vallin et al., 2005*; *Drinkwater et al., 2022*), presumably to emphasise the conspicuousness of the patterns. Most of the current literature does not take these effects into account in experiments, although some studies argue in favour or against their importance (e.g. *Blest, 1957a*; *Vallin et al., 2005*). We should also consider how factors other than those constituting the pattern (e.g. colour, number, and size) are involved in the predator avoidance function of eyespots. The location of the butterfly's eyespot patterns varies from species to species as well; eyespots exist on the wings' ventral, dorsal, or both sides. Not only the dorsal eyespot patterns, which were used in most studies, but also the ventral eyespot patterns should be explored. In addition, we need to avoid presenting patterns unnaturally when using real butterflies in experiments. For example, many owl butterflies (family Caligo) have a pair of eyespot patterns on the ventral side. Their eyespots are usually visible to birds when the wings are closed and would not present side by side as in the eyes of the owl's frontal face.

Third, recent studies have shown that birds are sensitive to the gaze of other individuals and may respond more aversively when their gazes are directed at them (e.g. *Carter et al., 2008*; *Clucas et al., 2013*; *Davidson et al., 2015*). *Skelhorn and Rowland, 2022* showed that the anti-predation effect may be further enhanced if the inner circle of the eyespot is in a more gazing-like position for subject birds. However, further research is needed to investigate the importance of the position of the inner circle.

Fourth, as mentioned above, studies focusing on caterpillar eyespots are much more scarce compared to butterflies; *Hossie and Sherratt, 2014* have shown similarities between caterpillars and snakes, but the response of birds to actual caterpillars has not been experimentally tested. Conversely, in butterflies, similarities between the eyespot patterns on wings and the eyes of birds of prey have not been investigated.

Finally, birds are generally considered as potential predators of butterflies and caterpillars. Although other taxa species, such as invertebrates (*Sang and Teder, 2011*; *Prudic et al., 2015*; *Chan et al., 2021*), lizards (*Lyytinen et al., 2003*; *Vlieger and Brakefield, 2007*; *Halali et al., 2019*), and rats (*Wiklund et al., 2008*; *Olofsson et al., 2011*; *Olofsson et al., 2012*; *Postema, 2022*), are also known to prey on lepidopterans, there are much fewer studies using non-avian species as predators. The effectiveness of eye mimicry versus being conspicuous may vary depending on the predator, and either one may be more effective depending on specific predator species. Therefore, we should expand the range of taxa used for experiments to get a better and more generalisable understanding of the eyespots' function and evolution in butterflies and caterpillars. Additionally, much of the research has been conducted in Europe and North America. Of the studies we included, only two were from other regions (India *Mukherjee and Kodandaramaiah, 2015* and Singapore *Ho et al., 2016*). The empirical results may differ in areas with many species of lepidopterans with eyespot patterns (e.g. *Ord et al., 2021*).

Knowing the effects of conspicuous patterns may contribute to creating a world where birds and humans can live more harmoniously. Both eyespots and non-eyespot patterns have already been used to control birds, particularly in agriculture, although their effectiveness has been questioned (e.g. *Avery et al., 1988*; *Nakamura et al., 1995*). Such uncertainty may reflect our limited understanding of why birds avoid eyespots and non-eyespots. Nevertheless, visual stimuli are less likely to harm birds

**Table 1.** Descriptions of the population, intervention, comparator, and outcome (PICO) were used to define the scope of this study.

| PICO | Description |
| --- | --- |
| Population | Birds as predators and butterflies, moths, caterpillars, and their models as prey |
| Intervention | Presenting eyespot or conspicuous pattern stimulus to birds |
| Comparator | Presenting stimulus that is neither eyespot nor conspicuous patterns |
| Outcome | Avian behavioural responses to eyespot or conspicuous pattern stimuli<br>The probability of prey surviving or being attacked (for the stimuli) |

or affect the natural environment than others (e.g. nest/egg destructions or toxic chemicals; reviewed in *Linz et al., 2015*). Therefore, when proven effective, they could be used for better pest control, population management, and conservation (*McLennan et al., 1995*).

## Conclusion

We have shed light on a traditional but controversial research topic that has fascinated behavioural ecologists for decades. Our findings provide a better understanding of the evolution of signal designs, but also show that more work is needed to understand the function of the eyespot patterns in Lepidoptera, such as whether eyespot patterns evolved due to mimicry or conspicuousness.

# Materials and methods

We preregistered our methods and planned analyses before data extraction and analysis in Open Science Framework (https://osf.io/ymwvb; *Mizuno et al., 2023*).

## Search protocols

We used the PICO (Population, Intervention, Comparator, Outcome; *Table 1*) framework (*Foo et al., 2021*) to specify the scope of our research questions and to inform our literature searching and screening. We conducted a comprehensive literature search across multiple databases, including Scopus, ISI Web of Science, Google Scholar (for non-English studies), and Bielefeld Academic Search Engine (for unpublished theses; i.e. grey literature). We designed the search strings (see *Supplementary file 4*) to identify studies that used experimental methods to examine the effects of eyespot patterns on birds' predation behaviours. We did not set any temporal restrictions on the database searches. Additionally, we conducted backward and forward reference searches within the Scopus database using four key publications (*Stevens, 2005*; *Kodandaramaiah, 2011*; *Stevens and Ruxton, 2014*; *Drinkwater et al., 2022*). The strings were translated for searches in non-English languages, and search results were assessed by reviewers with expertise in the respective languages: AM for Japanese, ML for Polish and Russian, PP for Portuguese and Spanish, and YY for Simplified and Traditional Chinese. We limited Google Scholar searches to the top 100 results in each language, sorted by relevance. In cases of disagreement between the reviewers, discrepancies were discussed and resolved to reach a consensus. The screening process and results are shown in the PRISMA-like flowchart (*Figure 2a*).

## Eligibility criteria

We set specific criteria for including studies in our meta-analysis (according to our pre-registered protocol). Initial screening, including titles, abstracts, and keyword assessment for English-language bibliographic records, was conducted by AM and ML using Rayyan (https://www.rayyan.ai; *Ouzzani et al., 2016*) following predefined inclusion criteria. Subsequently, AM and PP independently screened the full texts of studies that passed the initial screening. To be eligible, a study had to conduct experiments and provide data on bird behavioural responses or prey survival/attacked rates. We excluded studies solely involving non-avian predators, such as fish, insects, mammals, or other species. However, studies that included a mix of species from different taxonomic groups were allowed if the primary focus was on avian predation. In our analysis, we only considered research that presented both conspicuous and control (non-conspicuous) patterns as stimuli. We omitted studies using actual

predator or human eyes as stimuli since we focused on understanding how eyespot patterns in butterflies and caterpillars, which are unlikely to resemble specific bird or vertebrate species eyes, affect predation avoidance (*Janzen et al., 2010*). We also excluded studies that used bright and contrasting patterns as control stimuli because such stimuli would prevent comparison with eyespots or non-eyespot patterns. Furthermore, we focused only on studies that used real or artificial butterflies, moths, caterpillars, or a piece of paper as prey or presented stimuli. We also did not consider research that only investigated avian physiological responses to conspicuous patterns. In addition, we did not include studies that only assessed whether prey with eyespots or conspicuous patterns were less likely to be attacked by birds, based on wing or body damage alone, without including control stimuli. This is because it was not possible to quantitatively assess the effect of eyespots or non-eyespot patterns on predation avoidance without control stimuli.

## Data collection

We extracted four types of information from each study. First, we collected citation information, such as title, author name, and publication year. Second, we gathered the details of the presented stimuli used in each experiment within studies: type of control pattern (plain neutral-coloured or camouflaged), type of treatment pattern (eyespots or non-eyespot patterns), pattern area (mm$^2$: area per shape comprising the pattern), total pattern area (mm$^2$: when multiple patterns exist on the presented stimulus, it denotes the total area of all patterns; for stimuli with single eyespot or distinct pattern, the value equals the pattern area), linear size of the pattern (mm: e.g. maximum diameter or length of pattern), number of shapes in pattern, total area of prey surface (mm$^2$: e.g. butterfly wings and caterpillar bodies), prey material type (i.e. whether a real butterfly or a complete imitation of a particular butterfly was used as prey), and prey shape type (a further subdivision of the former). For non-eyespot patterns, we also noted pattern shapes (e.g. circles, stripes, and triangles). In each study, bird responses to control and treatment pattern stimuli and prey survival/attacked rates when these patterns were present were reported. Bird responses contained a variety of measures, including the number of attacks and escape behaviours, latency to attack, latency to approach, and the proportion of birds attacking the presented stimuli. Henceforth, we refer to these measures and responses as 'predator avoidance.' Third, we obtained data for calculating effect sizes (e.g. mean, standard deviation or standard error, and sample size of control and treatment group) from plots using WebPlotDigitizer 4.6.0 (https://automeris.io/WebPlotDigitizer), detailed tables, texts, or raw data. In survival analysis plots, we extracted data at the point in time when the difference between the 'survival' or 'attacked' rates of the intervention and comparison groups was greatest as outcomes. Study design (i.e. whether experiments were done independently or dependently between the control and treatment group) was also recorded. Fourth, we gathered predator and prey information, specifically, the study species (common English name and scientific name) and predator diet type. In some cases, studies did not use a specific bird species as a predator or a specific lepidopteran species as prey. We contacted authors when such information was ambiguous or missing. When the paper did not report the pattern area and diameter of the treatment stimulus or the presented stimulus surface area, AM calculated or measured them from available images using ImageJ v.1.53i (*Abramoff and Ram, 2004*).

The dataset was originally divided into two parts. The first part involved the data from presenting eyespot patterns to avian predators and directly observing their responses (predator dataset). The sample size or unit of analysis in this part was based on the number of individual avian predators. The second part involved the data from using real or artificial abstract butterflies, moths, or caterpillars with eyespots or non-eyespot patterns as stimuli or prey, and observing their survival/attacked probabilities in the field (prey dataset). The sample size or unit of analysis in this part was based on the number of real or artificial abstract prey. However, we also used the combined dataset that included both predator and prey datasets, as detailed in the '**Meta-analysis and meta-regressions'** and '**Publication bias'** sections.

## Effect size calculation

To obtain the effect size point estimates and sampling variances, we used lnRR (the natural logarithm of the response ratio) between the means of the treatment and the treatment control stimulus groups (*Hedges et al., 1999*; *Lajeunessei, 2011*; *Senior et al., 2020*). Positive lnRR values indicate heightened aversion in birds and enhanced prey survival, while negative lnRR values signify diminished bird

aversion and increased prey mortality. The point estimate and sampling variance (var) of lnRR can be then calculated in:

$$lnRR = ln\left(\frac{M_T}{M_C}\right) \tag{1}$$

$$var\left(lnRR\right) = \frac{SD_T^2}{N_T M_T^2} + \frac{SD_C^2}{N_C M_C^2} - 2r\sqrt{\frac{SD_T^2}{N_T M_T^2}}\sqrt{\frac{SD_C^2}{N_C M_C^2}} \tag{2}$$

where $M_T$ and $M_C$ are mean responses of treatment and control groups (e.g. total frequency of attacking prey, latency of approach, or prey survivability), respectively. $SD$ and $N$ are (sample) standard deviations and sample size, respectively. The term, $r$ is the correlation coefficient between responses of the two groups. Some of our eligible studies used the paired (dependent) study design where treatment and control samples originated from the same individuals, and sample sizes between the two groups were the same. None of these studies provided an estimate of $r$. Thus, when calculating our effect sizes, we assumed that this correlation was 0.5, which is conservative (*Noble et al., 2017*). For the other studies that used independent study design, we set $r = 0$.

We note that our dataset included proportion (percentage) data (e.g. predator attack rate or prey survival probability), which are bounded at 0 (0%) and 1 (100%). Therefore, we transformed group means ($M$) and group standard deviations ($SD$) for proportion data using *Equations (3) and (4)* before applying (1) and (2) to calculate lnRR and the sampling variance:

$$f\left(M\right) = arcsine\left(\sqrt{M}\right) \tag{3}$$

$$SD\left(f\left(M\right)\right) = \sqrt{\frac{SD^2}{4M\left(1 - M\right)}} \tag{4}$$

where $f$ indicates a function, in our case, the arcsine transformation. The standard deviation (SD) related to this transformation was derived using the delta method before calculating lnRR and the sampling variance (*Macartney et al., 2022*). We have also assumed that the standard deviation was $SD\left(f\left(M\right)\right) = 1/\sqrt{8}$ if SD was not available.

## Meta-analysis and meta-regressions

We used the *rma.mv* function from the package meta for v.4.4.0 (*Viechtbauer, 2010*) in R v.4.3.1 (*R Development Core Team, 2023*) for our analyses. We started by fitting multilevel, mixed-effect meta-analytic models to the predator and prey datasets. These meta-analytic models explicitly incorporated random factors, Study ID, Cohort ID (groups of the same subjects), and Shared control ID (indicating effect sizes sharing control groups) (*Nakagawa et al., 2023b*) along with Observation ID, fitted by the above function (*Viechtbauer, 2010*). The model for the predator dataset included Species ID and a correlation matrix related to phylogenetic relatedness for the species as random factors (*Nakagawa and Santos, 2012*). This is because we had data on the bird species used in the experiment in the predator dataset, and we needed to control for phylogenetic relationships between birds. We also quantified the total $I^2$ (a measure of heterogeneity not attributed to sampling error: *Higgins et al., 2003*) and how much each random factor was explained (partial $I^2$), calculated by the *i2_ml* function from the package orchaRd v.2.0.0 (*Nakagawa et al., 2023a*). After running both meta-analytical models, we found that phylogeny and Species ID did not need to be controlled for in the predator dataset, as their partial $I^2$ were zero ($I^2$=0.00%). That is, these factors explained little heterogeneity between effect sizes.

Therefore, we merged predator and prey datasets (i.e. full dataset) without considering phylogenetic information and used them for the following models. We had, as random effects, Study ID, Cohort ID, Shared control ID, and Observation ID for our meta-analytic model using the full dataset. The Cohort ID and Shared control ID were removed from our subsequent meta-regressions because they both explained little heterogeneity (both partial $I^2$<0.001%). This intercept-only (meta-analytic) model tested the conspicuous patterns (eyespots and non-eyespots) that affected predator avoidance (i.e. our first question).

Next, we tested whether eyespots and non-eyespot patterns differ in the magnitude and direction of the effect of elicited bird predator avoidance and what factors contribute to the deterring

effects of conspicuous patterns. We performed uni-moderator meta-regression models with each of eight moderators: treatment stimulus pattern types, pattern area, the number of pattern shapes, prey material type, maximum pattern diameter/length, total pattern area, total area of prey surface, and prey shape type (*Figure 2 bc*). We also ran a multi-moderator meta-regression model, including the first four of the eight variables mentioned in the uni-moderators, due to moderator correlations. We used log-transformed data for pattern area, total pattern area, total area of prey surface, and pattern maximum diameter/length in our analysis to normalise these moderators. We created all result plots in the *orchard_plot* and *bubble_plot* functions from the package orchaRd (*Nakagawa et al., 2023a*).

## Publication bias

We used three approaches to assess the presence of publication bias in our study. First, we visually assessed the funnel plot asymmetry by examining the residuals from a meta-analytic model, which included all the random factors utilised in our study. These residuals were plotted against the precision of the effect sizes. Second, we performed an alternative method to Egger's regression. This method used the inverse of the effective sample size as a moderator within a multilevel meta-analytic model (*Nakagawa et al., 2022*). Third, we examined the possibility of time-lag bias by including publication year as a moderator in our multilevel meta-analytic model. Uni-moderator models were run for each inverse of the effective sample size and publication year, and a multi-moderator model was carried out with the full model including both inverse of the effective sample size and publication year as moderators.

## Additions and deviations

We made two changes to the pre-registration: the addition of four new moderators and the removal of two moderators. The new moderators were pattern area, total pattern area, total area of prey surface, and prey shape types, although similar moderators were in the pre-registration such as the number of eyespots (patterns) and diameter of an eyespot (a pattern). These *post-hoc* decisions were taken to refine our initial moderators. We subsequently used them in our meta-regression analyses. We originally intended to include the broad outcome categories of predator avoidance measure as a moderator in the models, but the diversity of reported results made categorisation impossible. Therefore, we did not include it as a moderator. We also collected information on bird diet but decided not to include it. This decision was because six of the seven bird species in our study were omnivores, resulting in a lack of variability needed to detect diet effects in our data (for more details, please see **Results**).

## Acknowledgements

The authors thank Martin Stevens and Ben Brilot for sharing data. This work was supported by Japan Society for the Promotion of Science Research Fellowship for Young Scientists [JP22KJ0076], Japan Society for the Promotion of Science Overseas Challenge Program for Young Researchers [202280247] to Ayumi Mizuno; ARC [DP210100812, DP230101248] to Malgorzata Lagisz and Shinichi Nakagawa; Japan Society for the Promotion of Science Grant-in-Aid for Scientific Research [20K06809] to Masayo Soma.

## Additional information

### Funding

| Funder | Grant reference number | Author |
| --- | --- | --- |
| Japan Society for the Promotion of Science | JP22KJ0076 | Ayumi Mizuno |
| Japan Society for the Promotion of Science | 202280247 | Ayumi Mizuno |
| Australian Research Council | DP210100812 | Malgorzata Lagisz Shinichi Nakagawa |

| Funder | Grant reference number | Author |
| --- | --- | --- |
| Australian Research Council | DP230101248 | Malgorzata Lagisz<br>Shinichi Nakagawa |
| Japan Society for the Promotion of Science | 20K06809 | Masayo Soma |

The funders had no role in study design, data collection and interpretation, or the decision to submit the work for publication.

## Author contributions

Ayumi Mizuno, Conceptualization, Data curation, Software, Formal analysis, Supervision, Funding acquisition, Investigation, Visualization, Methodology, Writing - original draft, Project administration, Writing – review and editing; Malgorzata Lagisz, Data curation, Funding acquisition, Investigation, Methodology, Writing – review and editing; Pietro Pollo, Yefeng Yang, Data curation, Investigation, Writing – review and editing; Masayo Soma, Conceptualization, Funding acquisition, Visualization, Writing – review and editing; Shinichi Nakagawa, Conceptualization, Software, Supervision, Funding acquisition, Methodology, Project administration, Writing – review and editing

## Author ORCIDs

Ayumi Mizuno ⓘD https://orcid.org/0000-0003-0822-5637
Malgorzata Lagisz ⓘD https://orcid.org/0000-0002-3993-6127
Pietro Pollo ⓘD http://orcid.org/0000-0001-6555-5400
Yefeng Yang ⓘD https://orcid.org/0000-0002-8610-4016
Masayo Soma ⓘD http://orcid.org/0000-0002-8596-1956
Shinichi Nakagawa ⓘD https://orcid.org/0000-0002-7765-5182

Reviewer #1 (Public Review): https://doi.org/10.7554/eLife.96338.3.sa1
Author response https://doi.org/10.7554/eLife.96338.3.sa2

# Additional files

## Supplementary files

• Supplementary file 1. Preferred Reporting Items for Systematic Reviews and Meta-Analyses (PRISMA)-EcoEvo Checklist.

• Supplementary file 2. List of (a) included and (b) excluded studies at the full-text screening stage with exclusion reasons.

• Supplementary file 3. Summary of a multi-moderator model including all moderators. The bold typeface is used when a 95% confidence interval (CI) does not contain zero; thus, it can be interpreted as an existing significant effect in predator avoidance.

• Supplementary file 4. Search strings used for each database. We accessed Scopus, ISI Web of Science core collection, Google Scholar (*Japanese*, *Polish*, *Portuguese*, *Russian*, *Spanish*, *Simplified Chinese*, and *Traditional Chinese*) on 08/06/2023, and Bielefeld Academic Search Engine (BASE) on 26/06/2023. BASE was used as a source of grey literature. We conducted backward and forward reference searches for key review articles using Scopus on 19/06/2023. We modified search strings to collect studies to capture studies examining the effects of eyespot patterns on birds using experimental methods. Search strings were adapted to the structure of each database.

• Supplementary file 5. Average maximum diameter of eyespots on *Bicyclus anynana*. AM obtained the pictures from lepdata.org/photos/animals/ and https://data.nhm.ac.uk/ and measured the eyespot diameters. Raw data is available here: https://ayumi-495.github.io/eyespot/ and on GitHub (copy archived at *Mizuno, 2024*) and Zenodo.

• MDAR checklist

## Data availability

Raw data, analysis script and supplementary materials are available at https://ayumi-495.github.io/eyespot/ and GitHub (copy archived at *Mizuno, 2024*) and Zenodo.

The following dataset was generated:

| Author(s) | Year | Dataset title | Dataset URL | Database and Identifier |
|---|---|---|---|---|
| Mizuno A, Nakagawa S | 2024 | A systematic review and meta-analysis of eyespot anti-predator mechanisms | https://doi.org/10.5281/zenodo.13147019 | Zenodo, 10.5281/zenodo.13147019 |

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
