## [Editor Report · eLife assessment]

This meta-analysis presents **valuable** findings that reexamine the function of butterfly eyespots in predator avoidance and report for conspicuousness over mimicry. The analysis is robust, but the evidence supporting the importance of conspicuousness is **incomplete** due to the limitations of the literature, and this debate would benefit from additional experiments that would strengthen these claims. This paper is of interest to evolutionary biologists and ecologists working on the evolution of morphology and predator-prey interactions.

---

## [Referee Report · Reviewer #1 (Public Review)]

The question of whether eyespots mimic eyes has certainly been around for a very long time and led to a good deal of debate and contention. This isn't purely an issue of how eyespots work either, but more widely an example of the potential pitfalls of adopting 'just-so-stories' in biology before conducting the appropriate experiments. Recent years have seen a range of studies testing eye mimicry, often purporting to find evidence for or against it, and not always entirely objectively. Thus, the current study is very welcome, rigorously analysing the findings across a suite of papers based on evidence/effect sizes in a meta-analysis.

The work is very well conducted, robust, objective, and makes a range of valuable contributions and conclusions, with an extensive use of literature for the research. I have no issues with the analysis undertaken. The results and conclusions are compelling. It's probably fair to say that the topic needs more experiments to really reach firm conclusions but the authors do a good job of acknowledging this and highlighting where that future work would be best placed.

---

## [Author Response]

The following is the authors’ response to the original reviews.

**Public Reviews:**

**Reviewer #1 (Public Review):**
Summary:The question of whether eyespots mimic eyes has certainly been around for a very long time and led to a good deal of debate and contention. This isn't purely an issue of how eyespots work either, but more widely an example of the potential pitfalls of adopting 'just-so-stories' in biology before conducting the appropriate experiments. Recent years have seen a range of studies testing eye mimicry, often purporting to find evidence for or against it, and not always entirely objectively. Thus, the current study is very welcome, rigorously analysing the findings across a suite of papers based on evidence/effect sizes in a meta-analysis.Strengths:The work is very well conducted, robust, objective, and makes a range of valuable contributions and conclusions, with an extensive use of literature for the research. I have no issues with the analysis undertaken, just some minor comments on the manuscript. The results and conclusions are compelling. It's probably fair to say that the topic needs more experiments to really reach firm conclusions but the authors do a good job of acknowledging this and highlighting where that future work would be best placed.Weaknesses:There are few weaknesses in this work, just some minor amendments to the text for clarity and information.

We greatly appreciate Reviewer 1’s positive comments on our manuscript. We also revised our manuscript text and a figure in accordance with Reviewer 1’s recommendations.

**Reviewer #2 (Public Review):**
Many prey animals have eyespot-like markings (called eyespots) which have been shown in experiments to hinder predation. However, why eyespots are effective against predation has been debated. The authors attempt to use a meta-analytical approach to address the issue of whether eye-mimicry or conspicuousness makes eyespots effective against predation. They state that their results support the importance of conspicuousness. However, I am not convinced by this.There have been many experimental studies that have weighed in on the debate. Experiments have included manipulating target eyespot properties to make them more or less conspicuous, or to make them more or less similar to eyes. Each study has used its own set of protocols. Experiments have been done indoors with a single predator species, and outdoors where, presumably, a large number of predator species predated upon targets. The targets (i.e, prey with eyespot-like markings) have varied from simple triangular paper pieces with circles printed on them to real lepidopteran wings. Some studies have suggested that conspicuousness is important and eye-mimicry is ineffective, while other studies have suggested that more eye-like targets are better protected. Therefore, there is no consensus across experiments on the eye-mimicry versus conspicuousness debate.The authors enter the picture with their meta-analysis. The manuscript is well-written and easy to follow. The meta-analysis appears well-carried out, statistically. Their results suggest that conspicuousness is effective, while eye-mimicry is not. I am not convinced that their meta-analysis provides strong enough evidence for this conclusion. The studies that are part of the meta-analysis are varied in terms of protocols, and no single protocol is necessarily better than another. Support for conspicuousness has come primarily from one research group (as acknowledged by the authors), based on a particular set of protocols.Furthermore, although conspicuousness is amenable to being quantified, for e.g., using contrast or size of stimuli, assessment of 'similarity to eyes' is inherently subjective. Therefore, manipulation of 'similarity to eyes' in some studies may have been subtle enough that there was no effect.There are a few experiments that have indeed supported eye-mimicry. The results from experiments so far suggest that both eye-mimicry and conspicuousness are effective, possibly depending on the predator(s). Importantly, conspicuousness can benefit from eye-mimicry, while eye-mimicry can benefit from conspicuousness.Therefore, I argue that generalizing based on a meta-analysis of a small number of studies that conspicuousness is more important than eye-mimicry is not justified. To summarize, I am not convinced that the current study rules out the importance of eye-mimicry in the evolution of eyespots, although I agree with the authors that conspicuousness is important.

We understand Reviewer 2’s concerns and have addressed them by adding some sentences in the discussion part (L506- 508, L538-L540). In addition, our findings, which were guided by current knowledge, support the conspicuousness hypothesis, but we acknowledge the two hypotheses are not mutually exclusive (L110-112). We also do not reject the eye mimicry hypothesis. As we have demonstrated, there are still several gaps in the current literature and our understanding (L501-553). Our aim is for this research to stimulate further studies on this intriguing topic and to foster more fruitful discussions.

**Recommendations for the authors:**

**Reviewer #1 (Recommendations For The Authors):**
Minor commentsLines 59/60: "it is possible that eyespots do not involve mimicry of eyes..."

The sentence was revised (L59). To enhance readability, we have integrated Reviewer 1's suggestions by simplifying the relevant section instead of using the suggested sentence.

Line 61: not necessarily aposematism. They might work simply through neophobia, unfamiliarity, etc even without unprofitability

We changed the text in line with the comment from Reviewer 1 (L61-63).

Lines 62/63 - this is a little hard to follow because I think you really mean both studies of real lepidopterans as well as artificial targets. Need to explain a bit more clearly.

We provided an additional explanation of our included primary study type (L64-65).

Lines 93/94 - not quite that they have nothing to do with predator avoidance, but more that any subjective resemblance to eyes is coincidental, or simply as a result of those marking properties being more effective through conspicuousness in their own right.Line 94 - similarly, not just aposematism. You explain the possible reasons above on l92 as also being neophobia, etc.

We agreed with Reviewer 1’s comments and added more explanations about the conspicuousness hypothesis (L96-97). We have also rewritten the sentences that could be misleading to readers (L428).

Line 96 - this is perhaps a bit misleading as it seems to conflate mechanism and function. The eye mimicry vs conspicuousness debate is largely about how the so-called 'intimidation' function of eyespots works. That is, how eyespots prevent predators from attacking. The deflection hypothesis is a second function of eyespots, which might also work via consciousness or eye mimicry (e.g. if predators try to peck at 'eyes') but has been less central to the mimicry debate.

The explanations and suggestions from Reviewer 1 are very helpful. We revised this part of our manuscript (L103-108) and Figure 1 and its legend to make it clearer that the eyespot hypothesis and the conspicuousness hypothesis explain anti-predator functions from a different perspective than the deflection hypothesis.

There is a third function of eyespots too, that being as mate selection traits. Note that Figure 1 should also be altered to reflect these points.

We wanted to focus on explaining why eyespot patterns can contribute to prey survival. Therefore, we did not state that eyespot patterns function as mate selection traits in this paragraph. Alternatively, we have already mentioned this in the Discussion part (L455-L465) and rewrote it more clearly (L456).

Were there enough studies on non-avian predators to analyse in any way?

We found a few studies on non-avian predators (e.g. fish, invertebrates, or reptiles), but not enough to conduct a meta-analysis.

Line 171/72 - why? Can you explain, please.

The reason we excluded studies that used bright or contrasting patterns as control stimuli in our meta-analysis is to ensure comparability across primary studies. We added an explanation in the text (L180-181).

Line 177 - can you clarify this?

Without control stimuli, it is challenging to accurately assess the effect of eyespots or other conspicuous patterns on predation avoidance. Control stimuli allow for a comparison of the effect of eyespots or patterns. We added a more detailed explanation to clarify here (L186-188).

Line 309 - presumably you mean 33 papers, each of which may have multiple experiments? I might have missed it, but how many individual experiments in total?

There were 164 individual experiments. We have now added that information in the manuscript (L320).

Line 320 - paper shaped in a triangle mostly?

We cannot say that most artificial prey were triangular. After excluding the caterpillar type, 57.4% were triangular, while the remaining 43.6% were rectangular (Figure 2b).

Line 406: Stevens.

We fixed this name, thank you (L417).

Discussion - nice, balanced and thorough. Much of the work done has been in Northern Europe where eyespot species are less common. Perhaps things may differ in areas where eyespots are more prevalent.

We appreciate Reviewer 1’s kind words and comments. We agree with your comments and reflected them in our manuscript (L542-545).

Line 477 - True, and predators often have forward-facing eyes making it likely both would often be seen, but a pair of eyes may not be absolutely crucial to avoidance since sometimes a prey animal may only see one eye of a predator (e.g. if the other is occluded, or only one side of the head is visible).

We were grateful for Reviewer 1's comment. We added a sentence noting that the eyespots do not necessarily have to be in pairs to resemble eyes (L490-L492).